# Position: Comprehensive AI governance requires addressing non-model capability gains

**Arthur Goemans** [1]  **Dan Altman** [2]  **Noemi Dreksler** [3]  **Jonas Freund** [3]  **Milan Gandhi** [1]  **Zhengdong Wang** [1]
**Sarah Cogan** [1]  **Sebastien Krier** [1]  **Demetra Brady** [1]  **Lewis Ho** [1]  **Allan Dafoe** [1]

## Abstract

Frontier AI governance often centres on the model-level governance paradigm, which assumes that a model's capability profile is primarily a function of the compute and data used during training. This position paper argues that model-level governance becomes less effective when capability progress is increasingly driven by "non-model gains"—improvements that are independent from advances in the base model. We formalise the concept of non-model gains and provide a taxonomy of three distinct vectors of capability gain: *inference gain* (scaling compute at test-time), *systems gain* (post-training enhancements such as scaffolds), and *asset gain* (enhancing a model with restricted assets). We demonstrate how these vectors—alongside potential future impacts from embodiment, continual learning, and AI diffusion—may undermine risk management strategies that hinge mostly on pre-deployment evaluation and mitigation. We provide an overview of governance approaches that go beyond the model level: system, entity, agent, and cloud governance. Finally, we emphasise the importance of societal resilience as a complement to these governance layers.

## 1. Introduction

What are the limits and alternatives to model-level governance approaches? We offer an assessment to inform policy debates on AI governance and prepare the AI ecosystem for a transforming risk landscape. Below is an overview of the paper structure and content.

***Section 1: Limits to model-level governance.*** Model-level governance assumes that a model's capability profile is primarily a function of the compute and data used in the training process. Model-level governance becomes less effective when capability progress is increasingly driven by non-model gains—improvements independent from advances in the base model.

***Section 2: Analysis of non-model gains.*** We identify three distinct vectors of capability gain: *inference gain* (scaling compute at test-time), *systems gain* (post-training enhancements such as scaffolds), and *asset gain* (access to restricted assets such as biological data or military hardware). We also address non-model gains that may become prominent in the future: *embodiment (*the use of models in physical embodiments*)*, *continual learning* (the ability of models to evolve over time), and *diffusion* (emergent effects from wide-scale deployment of AI). Each type of non-model gain impacts the model-level governance paradigm, thereby weakening a risk management strategy based mostly on pre-deployment evaluation and mitigation of frontier model risks.

***Section 3: Governance strategies.*** Model-level governance can partially adapt to non-model gains by focusing on an evolutive elicitation standard, partnerships focused on improving the elicitation stack, post-deployment monitoring, and forecasting capability overhang. Furthermore, we should consider governance approaches beyond the model level: *system governance* (governing the systems built atop frontier models), *entity governance* (governing the organisations that provide frontier models or systems), *agent governance* (governing delegation to and interaction between AI agents), and *cloud governance* (governing the cloud deployment of frontier models and systems). Societal resilience serves as a complement to the above governance patchwork.

***Section 4: Alternative views.***

*What if advances in base model capabilities remain the primary constraint on performance?* We do not dispute that model-level governance remains relevant; our argument is that it should be complemented rather than replaced, and that the relative returns to model-level intervention may diminish as non-model gains grow in significance.

---

[1]Google DeepMind [2]Work completed while at Google DeepMind. Currently at Anthropic. [3]Centre for the Governance of AI. Correspondence to: Arthur Goemans <arthur.goemans@gmail.com>.

*Proceedings of the $43^{rd}$ International Conference on Machine Learning*, Seoul, South Korea. PMLR 306, 2026. Copyright 2026 by the author(s).

*Isn't it true that many of your alternative governance strategies face prohibitive practical, legal, and technical barriers?* There are certainly implementation challenges, but the prominence of non-model gains necessitates a broader governance portfolio. While some of these challenges may prove insurmountable, a comprehensive analysis should happen sooner rather than later.

***Section 5: Call to action.*** We recommend establishing better metrics for non-model gains and an exploration of post-deployment monitoring and invest in forecasting methodologies to track the evolving risk landscape. We also call for the technical and research community to evaluate governance mechanisms that go beyond model-level and focus on system, entity, agent, and cloud nodes.

## 2. Model-level Governance and its Limits

**Much of the capability progress for large language models (LLMs) has been driven by scaling compute and data in the training process.** Scaling pre-training has yielded predictable performance results, illustrated by scaling laws that describe the power-law relationship between compute and performance (Hoffman et al., 2022; Henighan et al., 2020; Hernandez et al., 2021; Kaplan et al., 2020; Ruan et al., 2024; Hestness et al., 2017; Bahri et al., 2021). Since late 2024, the rise of reasoning models and synthetic data pipelines has turned post-training into an increasingly critical and resource-intensive phase (DeepSeek-AI, 2025a; Kumar et al., 2025; Lai et al., 2025; xAI, 2025).

**Due to the immense resource requirements for frontier training runs, model development typically takes place within a small number of companies.** These organisations have become the primary node for safety efforts. Accordingly, frontier model developers have implemented risk management frameworks that focus on assessing and mitigating risks prior to model deployment (METR, 2023; Anthropic, 2023; Google DeepMind, 2025a; OpenAI, 2025c). Similarly, many regulatory efforts have focused specifically on model development and deployment decisions by frontier model developers, often using training compute thresholds as a regulatory filter (European Union, 2024; European Commission, 2025a; New York State Senate, 2025; California State Legislature, 2025).

**We use the term model-level governance to describe governance measures informed by an AI model's capability profile.** Examples are the use of dangerous capability evaluations to assess downstream risks of a model, or applying security measures (e.g. access controls for model weights) or safety guardrails to the model (e.g. unlearning, behavioural alignment) to mitigate such risks (Frontier Model Forum, 2025; Nevo et al., 2024; Phuong et al., 2024; Ji et al., 2023; Yao et al., 2023b). Much of model-level governance to

date has focused on *pre-deployment* interventions, though more recently norms are emerging for *post-deployment* interventions, such as the post-market monitoring required by the EU AI Act Code of Practice (European Commission, 2025a). Not all AI governance measures proposed or in effect can be characterised as model-level. Examples of non-model AI governance measures include export controls for GPUs (U.S. Bureau of Industry and Security, 2024; Sastry et al., 2024) and whistleblower protections (European Commission, 2025b; California State Legislature, 2025).

**The effectiveness of model-level governance largely depends on the ability of frontier model developers to evaluate and mitigate dangerous capabilities before a model is released**. Conversely, model-level governance deteriorates when (1) it becomes harder for frontier model developers to elicit the downstream capabilities of their models ("elicitation failure"), (2) it becomes harder for frontier model developers to mitigate the dangerous capabilities they elicit ("mitigation failure"), or (3) the expected harm of failing to elicit a particular capability increases ("overhang cost").

**In this paper, we will focus on the first governance failure mode, arguing that pre-deployment elicitation alone may become structurally inefficient**. This is due to so-called "non-model gains": capability improvements that do not stem from pre-training or standard post-training techniques. This includes *inference gain* (capability improvements from scaling compute at inference time), *systems gain* (capability improvements associated with bespoke post-training enhancements, such as fine-tuning for a narrow use case, scaffolding, or tool use), and *asset gain* (capability improvements associated with access to restricted assets such as government-held expertise or classified data). We also address potential future non-model gains from *embodiment (*using models in physical embodiments*), continual learning* (the ability of models to evolve over time), and *diffusion* (emergent effects from wide-scale AI deployment).

**Non-model gains are not a new phenomenon, but they are becoming more salient as scaling pre-training compute alone accounts for less of the overall capability gain** (Ord, 2025; Somala et al., 2025). For example, inference scaling has been an important factor in the success of the reasoning models that first emerged late 2024, after reports of a scaling wall first gained traction (Ord, 2025). Additionally, non-model gains have become more significant now that base models are more advanced, enabling more complex orchestrations (e.g. multi-agent scaffolds and sophisticated tool use) and interactions once deployed.

**Non-model gains are highly accessible relative to the prohibitive cost of pre-training.** Thus, while frontier model developers typically try to anticipate non-model gains as part of the elicitation process (METR, 2024), they cannot predict all the ways in which their models will be used "in

the wild" (Poetiq, 2025; Anthropic, 2026; Brand & Denain, 2025).

**Therefore, model-level governance may need to be complemented with other governance approaches.** We imagine that current model-level efforts, whether based on voluntary commitments or regulatory requirements, will remain important in the coming years. We also believe that model-level governance can partially adapt to this new landscape. However, even if model-level governance evolves, non-model gains may increase AI risk if we do not diversify our governance portfolio.

## 3. Analysis of Non-Model Gains

**This section provides an overview of each type of non-model gain and how they undermine model-level governance.** While we provide a siloed overview here, a lot of capability breakthroughs emerge as a result of a combination of non-model gains rather than a singular type. For example, an AI cyber agent outperforms its base model through a combination of system gains to understand how to detect vulnerabilities, and inference gains to scan across many databases and test many exploits.

**We conclude each section with a reference to a potential governance complement.** These governance models are discussed in more detail in the next section.

### 3.1. Inference Gain

**Inference gain refers to the capability improvements resulting from scaling computational resources at inference time**. A pivot point was the advent of OpenAI's o1 series, models which displayed significant improvements through sequential and parallel inference scaling (OpenAI, 2024). Part of the reason why thinking models did not emerge earlier was that scaling inference on earlier models was not very effective (Wei et al., 2022; Weng et al., 2023). Reinforcement learning made models more amenable to inference scaling (Epoch, 2025), decreasing the relative importance of pre-training.

**Through inference scaling, smaller models can emulate the capabilities of larger models**. Within limits, one can offset less pre-training compute with more inference compute (Erdil, 2024) . This can shrink the gap between 'frontier' and 'sub-frontier' models: For example, a novel inference scaling technique called recursive self-aggregation enabled Qwen3-4B-Instruct-2507 to perform on par with larger reasoning models such as o3-mini (high) (Venkatraman et al., 2025). Additionally, DeepSeek-V3.2 outperformed the generally superior Gemini 3 on a series of prominent benchmarks by using between 1.5 and 2.5 times more tokens (DeepSeek-AI, 2025b) .

**To a large extent, model-level governance requires that powerful proprietary models maintain a lead over open-weight models, which are less amenable to mitigation.** As such, it is problematic when performance differences can be overcome through scaling inference. Over time, models may become even better at longer thought chains without loss of coherence, or may be robustly able to self-verify their responses. This would increase the room for inference scaling and further shrink the performance delta between frontier and sub-frontier models.

**The rise of inference gains undermines model-level governance.** The relative importance of inference versus pre-training has shifted since the advent of reasoning models. Therefore, malicious actors who manage to jailbreak a proprietary model may elicit unforeseen capabilities. Alternatively, because inference scaling shrinks the gap between proprietary and open weight models, malicious actors may turn to widely available open-weight models, bypassing the need for jailbreaking proprietary models. As argued below, this could be mitigated through changes in model-level governance in combination with non-model approaches such as entity and cloud governance.

### 3.2. Systems Gain

**Systems gain refers to the capability improvements from various enhancements applied after pre-training and standard post-training.** This includes tools, elaborate prompt and context engineering strategies, and multi-agent scaffolds or iterative self-improvement schemes (Davidson et al., 2023).

**The bulk of AI systems built atop frontier models do not advance the capability frontier in a meaningful way.** Many system enhancements (e.g. standard tool use and prompting schemes) have become ubiquitous—they do not alter the safety calculus of the base model.

**However, a subset of AI systems pushes the capability frontier.** A salient example is Google DeepMind's Big Sleep—a system integrating an LLM with tools like a code browser and debugger—which was the first LLM agent that discovered a zero day (Google Project Zero, 2024). While Big Sleep was engineered by a model developer, there are many third parties who have succeeded in building LLM-based systems that support automated vulnerability detection (DarkNavy, 2024; Wu et al., 2025; Sun et al., 2024; Du et al., 2024). Generally speaking, a model's performance hinges strongly on finding the best scaffold, which the developer might not always find prior to deployment (Poetiq, 2025; Brand & Denain, 2025).

**Advances in base model quality may serve as a force multiplier for system gains.** While basic system affordances yield diminishing returns as models become more

broadly competent, complex systems—such as multi-turn agentic orchestrations with extensive tool access—can exhibit increasing returns when paired with highly competent base models. This creates options for bad actors who want to misuse LLMs by integrating them in adversarial systems. The complex Claude Code scaffold built by a Chinese state-sponsored group in the fall of 2025 is a case in point (Anthropic, 2026).

**The rise of system gains undermines model-level governance.** Compared to pre-training scaling, system gains often come at a low cost. Complicated scaffoldings require expertise and access to inference compute to build and test, but above we cite evidence of low-resourced actors succeeding at these tasks. Furthermore, once the recipe for a scaffold is known, it proliferates freely—again, in contrast with a proprietary model. Finally, as better base models enable more system gains, frontier model developers will increasingly struggle to anticipate all possible downstream modifications of their models. As argued below, these risks can be mitigated through changes to model-level governance in combination with alternative approaches such as entity, agent, and system governance.

### 3.3. Asset Gain

**Asset gain refers to capability increases that result from combining an AI model with special assets not available to the original model developers or testers.** These assets can include government-held expertise (e.g., advanced propulsion, stealth technology, nuclear physics), specialised hardware (e.g. physical embodiments, sensors, actuators), or classified data (e.g. undisclosed software vulnerabilities, data on non-public or enhanced pathogens, or sovereign decryption keys). For instance, using the model with classified weaponisation protocols or within a sophisticated, government-only cyber range could unlock dangerous CBRN or cyber capabilities.

**Given the increasing significance of AI models for national security, we assume governments will increasingly consider integrating their exclusive strategic assets with AI models.** We find evidence of this in the uptick in partnerships between model developers and national security institutions (Anthropic, 2025b;c; Singh, 2025; Anduril Industries, 2024).

**Asset gain may undermine model-level governance, though it is hard to assess to which degree.** The restricted nature of these assets makes it very hard to estimate the magnitude of potential capability gains. In principle, the results could be significant—empirical research illustrates how specialised data sets can yield performance improvements equivalent to scaling pre-training compute by a factor of 1000 (Davidson et al., 2023).

**Identifying the right governance measures is challenging.** Unlike systems and inference gain, asset gain only undermines model-level governance with regard to the limited number of actors who hold such restricted assets. One solution is to conduct post-deployment monitoring or to work with national security authorities to improve elicitation stacks (see 4.1), though generalising results from such collaborations is difficult due to the idiosyncratic nature of restricted assets.

### 3.4. Looking Ahead: Other Limits to Model-Level Governance

Other limits to model-level governance may emerge in the years to come. We highlight three: embodiment, continual learning, and AI diffusion. We explain how they may further undermine the model-level governance paradigm.

#### 3.4.1. EMBODIMENT GAIN

**Embodiment gain refers to the capability improvements resulting from the integration of an AI model with physical actuators and sensors.** While traditional robotics relied on hard-coded control systems, recent advances in integrating multi-modal LLMs in robotic embodiments (Boston Dynamics, 2024; Addlesee et al., 2024; Spitale et al., 2023) and building specialist Vision-Language-Action (VLA) robotics models (Gemini Robotics Team, 2025; OpenVLA Team, 2024) allow frontier models to serve as the "brain" for general-purpose robots, translating semantic reasoning into kinetic action (Gemini Robotics Team, 2025; NVIDIA, 2024; Li et al., 2023; Driess et al., 2023; Kawaharazuka et al., 2025). Using transformer-based frontier models like LLMs or VLAs in robotics has the advantage of enabling generalisation and adaptability in robotic software, moving beyond static pre-programming to learning new physical tasks and environments without specific retraining.

**Through embodiment gain, the capabilities of a model shift from merely informational to physical.** Consequently, the risk profile of a frontier model may change when it is embodied. Concerns about a model's susceptibility to hallucinations and jailbreaks may translate into physical safety risks (Robey et al., 2024; Cohen et al., 2024). Managing such risks requires a different approach than the current model evaluation and mitigation practices—consistent with what we argue for AI systems, a governance solution would involve alignment between multiple actors in the supply chain (see 4.2.1).

#### 3.4.2. CONTINUAL LEARNING

**Continual learning refers to the ability of an AI model to incrementally acquire and exploit knowledge throughout its lifecycle** (Shi et al., 2025; Wang et al., 2023). Typically, continual learning requires that the model parameters

evolve over time. This often results in catastrophic forgetting, where learning new tasks reduces proficiency in old tasks (Shi et al., 2025), which is why the current generation of LLMs is largely static (Behrouz et al., 2025).

**Advances in continual learning could further undermine the model-level governance paradigm.** Continual learning could unlock greater capabilities in gaining up-to-date knowledge, adapting to dynamic situations, and personalisation. As such, a previously safe model could drift into unsafe behaviours, either through deliberate interventions from malicious users (e.g. using data poisoning) or through accidental unlearning of safety training. Post-deployment monitoring could be a strategy to alleviate these risks (see 4.1), though much will depend on how continual learning is operationalised.

### 3.4.3. DIFFUSION EFFECTS

**By diffusion effects, we refer to the aggregate effects of widespread deployment of and interactions between AI models and systems.** Diffusion can engender a 'collective capability' that cannot be predicted at the model level. Examples of risks associated with diffusion are *monoculture* (one or a few AI models may become dominant, so that subtle flaws or harmful propensities become ubiquitous (Kleinberg & Raghavan, 2021; Bommasani et al., 2022)) or *cascading failures* (when the ubiquity of AI agents in interdependent systems such as financial markets lead to snowball effects).

**As AI systems become more capable, they will be deployed more widely, exacerbating diffusion effects.** The degree to which this has adverse consequences depends on factors such as sectoral regulatory markets and societal resilience (see 4.2.5). Model-level governance would remain useful to govern certain dangerous capabilities, but a broader suite of governance mechanisms would be required to manage diffusion risks, such as entity and agent governance (see 4.2.2 and 4.2.3).

## 4. Governance Strategies

**In this section, we discuss governance strategies that can compensate for the limitations of model-level governance.** We first briefly cover how model-level governance approaches can evolve, before providing an overview of governance options outside of the model-level governance paradigm. Select elements of these approaches may already be customary, but overall these governance alternatives are less prominent in AI governance practice.

**In Table 1, we attempt to map how different approaches may mitigate the risks associated with non-model gains.** We reiterate that, while an exploration of different governance approaches is necessary, we do not expect model-level

governance to become redundant.

### 4.1. Enhanced Model-Level Governance

**Elicitation standards.** It becomes more challenging to estimate the capability ceiling of a model as non-model gains increase, for instance, because of resource demands for inference compute and scaffolding experiments. Rather than the absolute capability ceiling—an increasingly elusive target—the elicitation standard could be informed by rigorous threat modelling. Instead of "what is the model capable of", model developers are increasingly focusing on "what capabilities can an actor with given resources elicit over a given timespan?" However, even with an adaptive elicitation standard it might not be feasible for the model developer to anticipate the full range of capabilities that downstream actors might elicit.

**Partnerships.** To build more sophisticated and reliable elicitation stacks, developers can deepen cooperation with each other, with third-party testers, or with authorities. For instance, Anthropic and OpenAI have collaborated on alignment evaluations (Bowman et al., 2025). Developers could also engage in partnerships with AISIs and other third party evaluators to improve safeguards (OpenAI, 2025b; Anthropic, 2025a; OpenAI, 2025a) or work with national security authorities to build evaluations incorporating restricted assets or bespoke scaffolding. To understand diffusion effects, partnerships might need to include social scientists, economists, and platform companies. For embodiment, working together with hardware manufacturers can help to understand use cases and risks, adapting elicitation procedures accordingly.

**Post-deployment monitoring.** Evaluations could become continuous, re-assessing deployed models when new techniques around non-model gains emerge (Berez, 2026). To this effect, model developers could monitor leaderboards of LLM agent benchmarks (such as GAIA, AgentBench, SWE-Bench, MLE-bench, or Cybench) and other channels (e.g. X, Reddit, GitHub, ArXiv). Beyond public signals, developers can work with third parties to monitor downstream use and engage in threat intelligence sharing platforms to maintain awareness of the adversarial landscape (Goemans et al., 2024).

**Forecasting.** If evaluators become less effective at eliciting the maximum capabilities of a model, predicting the size of the capability overhang becomes more important. Currently, it is difficult to make forecasting load-bearing for frontier safety. However, frontier model developers such as Anthropic and Google DeepMind are investing in forecasting methodology and capacity (Phuong et al., 2024; Jones et al., 2025). Forecasting seems especially promising for inference gains, due to emerging scaling laws (Bian et al., 2025; Wu et al., 2024). System gains are less amenable to extrapo-

*Table 1.* Summary of governance strategies. Green boxes indicate preliminary usefulness of a governance strategy (horizontal) for a capability paradigm (vertical).

| CAPABILITY PARADIGM | ENHANCED MODEL-LEVEL GOVERNANCE | | | | BEYOND MODEL-LEVEL GOVERNANCE | | | | |
| --- | --- | --- | --- | --- | --- | --- | --- | --- | --- |
| | ELICITATION STANDARD | PARTNER-SHIPS | POST-DEPLOY. MONITORING | FORE-CASTING | SYSTEM GOV. | ENTITY GOV. | AGENT GOV. | CLOUD GOV. | SOCIETAL RESILIENCE |
| INFERENCE | ■ | | ■ | ■ | | ■ | | ■ | ■ |
| SYSTEMS | ■ | ■ | ■ | ■ | ■ | ■ | ■ | | ■ |
| RESTRICTED ASSETS | | ■ | ■ | | | | | | ■ |
| EMBODIMENT | | ■ | | | ■ | | | | ■ |
| CONTINUAL LEARNING | | | ■ | | | | | | ■ |
| DIFFUSION | | | ■ | | | ■ | ■ | | ■ |

lation, though it is possible to conduct empirical work on the performance gains from various system enhancements (Davidson et al., 2023). Forecasting seems less tractable for asset gain and diffusion effects, where even establishing a baseline ("nowcasting") is difficult.

## 4.2. Beyond Model-Level Governance

Even with the changes outlined in the previous section, model-level governance may need to be complemented with other governance approaches to manage AI risk. We discuss these approaches below.

### 4.2.1. SYSTEM GOVERNANCE

**Downstream actors use foundation models to develop vastly different systems, ranging from language learning apps, legal assistants, and customer service agents.** As explained above, many such AI systems do not meaningfully advance the capability frontier. It may not be feasible or useful to require these providers to engage in frontier safety practices, at least not beyond what is required by existing laws.

**When system integration provides significant uplift in the model's capabilities, system providers become an important node in frontier safety.** An example might include a third party coding assistant or general-purpose agent that is significantly more capable than its base model. In such a case, the system provider is best placed to take at least some responsibility for risk management (Somala et al., 2025). Often, this is already required by laws regarding product liability, tort, data protection, consumer protection, or sector-specific regulatory frameworks. Additionally, providers of those frontier systems could conduct evaluations and implement mitigations, akin to what many model developers do today. This is especially true for AI systems optimised for use in critical areas such as coding, ML R&D, or cybersecurity. However, identifying the relevant players from the large number of system providers is a challenging task.

**As downstream systems proliferate, it remains important that model developers assess the risk of their models—including risks associated with standard scaffold-ing and inference scaling.** In the cases where technically advanced system-level governance is required to mitigate downstream risk, frontier model developers can facilitate the work of the system provider, e.g. by issuing detailed model cards and conveying the purpose and limitations of evaluations that were performed at the model-level.

### 4.2.2. ENTITY GOVERNANCE

**Entity governance shifts attention from individual AI models or systems to the organisations that develop and deploy them** (Ball & Ramakrishnan, 2025). This reflects the view that an organisation's structure, incentives, culture, and decision-making processes play a central role in shaping how AI risks are handled in practice (Brundage et al., 2026). Specifically, entity governance includes risk internal policies (e.g. reporting channels, accountability mechanisms, transparency), structures (e.g. committees, councils, and dedicated safety or assurance functions), and other measures that impact organisational risk culture (ISO/IEC, 2023; Committee of Sponsoring Organizations of the Treadway Commission, 2017). More stringent approaches include registration, or authorisation requirements for certain entities, which could establish baseline expectations around safety capacity and responsible practice.

**Entity governance helps ensure that non-model gains are identified and analysed coherently, rather than addressed piecemeal.** For example, for inference gains, this might involve policies requiring additional review when models are deployed with extended reasoning capabilities. For system gains, governance mechanisms can ensure that novel scaffolds unlocking unexpected capabilities receive sufficient evaluation for misuse potential prior to deployment.

**Entity governance is best understood as a foundational layer rather than a standalone solution.** It helps anchor accountability, build internal risk management capacity, and align organisational incentives with safe development and deployment. However, entity governance has important limitations: first, it struggles to address risks arising from open-weight models, decentralised development, or diffuse ecosystems where responsibility is fragmented. Second,

entity governance in the form of licensing or entry requirements may privilege incumbents at the expense of competition and innovation. Third, there is a risk that organisational governance devolves into formalistic compliance. Therefore, entity governance works best when complemented by more targeted approaches—such as system or cloud governance.

### 4.2.3. AGENT GOVERNANCE

**Agent governance shifts the focus from the capabilities of an AI model or system to the parameters of delegation and autonomous interaction.** It aims to address some of the governance challenges that arise when AI agents are deployed at scale, make decisions without supervision, and interact with each other.

**Frontier safety frameworks typically address some of the dangerous capabilities of increasingly agentic models, though multi-agent dynamics may lead to system gains and diffusion effects that are not adequately covered.**. As agents become more advanced or execute complex workflows without supervision, they may introduce failure modes such as miscoordination, conflict, and collusion that cannot be assessed by evaluating an AI model in isolation (Hammond et al., 2025).

**Governance of risks from multi-agents dynamics may first and foremost benefit from better multi-agent evaluations**, which would provide a better understanding of dangerous capabilities or risks that arise from multi-agent collusion or miscoordination (Hammond et al., 2025).

**Additionally, agent infrastructure can reduce some of the risk associated with systems gain and diffusion effects.** This includes measures aimed at managing agent behavior (access boundaries, behavioral constraints, deployment restrictions), attribution (e.g. unique agent IDs), and interaction (e.g. protocols for inter-agent communication) (Chan et al., 2025; Google DeepMind, 2025b).

### 4.2.4. CLOUD GOVERNANCE

**If inference scaling drives capability gains, privacy-safe cloud-level monitoring could potentially be useful, though there are major hurdles**. In theory, there are various ways to monitor for dangerous capability misuse at the cloud level (Heim et al., 2024a):

- **Know Your Customer (KYC)** for cloud providers has been proposed before (Heim et al., 2024b), but deemed incompatible with other regulatory regimes, such as those focused on privacy (Information Technology and Innovation Foundation, 2024).

- **Content-based monitoring** could be achieved e.g. through automated monitoring of a targeted subset of cloud users' activities for indications of misuse,

leveraging technical methods to maintain users' privacy (Sastry et al., 2024; Trask et al., 2020). For customers using models served under Platform-as-a-Service (PaaS) frameworks, this could be done using an automated monitoring pipeline, designed to retain and inspect specific logs only when a high-precision qualifier indicates particular kinds of misuse (e.g. pathogen development). For customers renting hardware to run their own inference or PaaS customers with highly confidential workloads, content-based monitoring might require advances in confidential computing (Trask et al., 2020; Confidential Computing Consortium, 2023). Even with such techniques, it is questionable whether the required scale of the monitoring would be legally and commercially viable.

- **Monitoring computational patterns** means automated monitoring inference compute usage for anomalies. In principle, this could work for tasks that are (1) misuse-prone and (2) benefit from extensive inference scaling, such as using AI to identify vulnerabilities in code. In practice, it's uncertain whether the level of inference scaling required for such misuse is (and remains) exceptional enough to stand out to a cloud provider.

**These techniques look most promising when cumulated.** For example, a signal from the classifier about potentially dangerous content in combination with an inference compute spike could be a justification to inspect targeted raw logs if this can be done in a privacy-safe manner—unless the usage is consistent with the legitimate purposes of a verified user.

**Overall, the commercial, technical, and legal feasibility of cloud-level governance has yet to be investigated in depth**. Several of the challenges include contractual and legal privacy requirements, coordination between cloud providers, technical advances in confidential computing, and overcoming significant overhead associated with content-based monitoring.

### 4.2.5. SOCIETAL RESILIENCE

**While the governance layers described above—system, entity, agent, and cloud—aim to reduce the likelihood of risks materialising, they cannot eliminate them entirely.** As non-model gains increase access to advanced capabilities, governance alternatives may prove inadequate or require significant time to mature. Therefore, societal resilience—interventions aimed at helping communities to cope with and recover from external disturbances—can complement technical efforts to protect communities against risks that other governance mechanisms cannot capture (Bengio et al., 2025; Gandhi et al., 2025; Bernardi et al., 2025).

**Societal resilience interventions are important because powerful AI systems, amplified by the vectors discussed in this paper, will impact many different aspects of society**. One approach is to hypothesise interventions that would hinder malicious actors from weaponising frontier AI to inflict catastrophic damage (Bernardi et al., 2025). For instance, where asset gains allow actors to combine models with sensitive biological data, resilience measures can target the physical supply chain. Indeed, the US AI Action Plan mandates that federally funded researchers use nucleic acid synthesis providers with robust screening protocols, thereby adding a layer of defense against AI-enabled biological threats (The White House, 2025).

**Several promising field-building initiatives are underway that align with this logic.** For example, the Singapore Consensus (2025) identifies a need for research into strengthening economic and security infrastructure against AI-enabled disruption (Bengio et al., 2025). The UK AI Security Institute has launched a societal resilience programme, which involves studying real-world AI deployment to better understand risks, such as the financial instability that may emerge through the actions of autonomous agents (i.e. diffusion effects) (Bernardi et al., 2025). In parallel, developing privacy-preserving mechanisms to share statistical insights into model usage will be important. Understanding how AI is diffusing can enable evidence-based prioritisation of investments into societal resilience.

## 5. Alternative Views

This paper's central claim—that model-level governance loses effectiveness as capability gains are increasingly decoupled from scaling base models—faces at least two objections.

**Non-model gains do not meaningfully change the risk calculus, as base model capabilities still constrain what scaffolding or inference scaling can achieve.** Indeed, many high-profile demonstrations of substantial systems gains, such as Big Sleep and widely cited agentic frameworks, have been demonstrated using frontier or near-frontier models (Google Project Zero, 2024; Yao et al., 2023a; Park et al., 2023), meaning the model remains the key leverage point. However, we argue that, as the floor of accessible capabilities rises, the space of actors who can achieve dangerous outcomes expands—even if the absolute ceiling remains at the frontier. Our argument is that it should be complemented rather than replaced, and that the relative returns to model-level intervention may diminish as non-model gains grow in significance.

**Governance alternatives are premature or infeasible.** Entity governance requirements can be burdensome and devolve into box-ticking exercises. System governance must contend with the sheer number and heterogeneity of downstream actors. Cloud governance faces significant legal, technical, and commercial hurdles that we acknowledge in the paper. Critics may argue that diversifying governance prematurely spreads resources thin and creates an inefficient, fragmented patchwork of governance approaches. We do not claim that all proposed approaches are immediately implementable. Our argument is that preparing governance mechanisms beyond the model level is prudent given the trajectory of non-model gains. Waiting instead until model-level governance has demonstrably failed risks leaving the governance ecosystem unprepared.

## 6. Call to Action

Our analysis of the limits of model-level governance and the rise of non-model gains suggests the following calls to action:

**Better understand and measure non-model gain**. The research community, industry, and AISIs should invest more in understanding how inference, systems, and asset gain affect AI system's ability to cross dangerous capability thresholds. This will build up more established science on how to evaluate the current impact as well as the trajectory of different non-model gains.

**Explore continuous post-deployment monitoring**. Model developers and deployers may consider monitoring public channels (Github, arXiv, social media) and agent benchmarks to spot when the community discovers new ways to boost dangerous model capabilities. They can take this information into account to recalibrate risk management efforts.

**Invest in forecasting methodologies**. The research community and industry should advance methodologies for forecasting capability overhang. Researchers should focus on nowcasting and forecasting the performance delta between base models and their enhanced states (i.e. via inference gains or systems gains) to provide stakeholders with leading indicators of when models may cross critical risk thresholds.

**Develop complementary governance mechanisms**. The technical and academic community should research the feasibility of governance mechanisms mentioned by this paper which complement model-level governance. These include system, entity, agent, and cloud governance mechanisms.

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
