# OpenReview forum: "Position: Comprehensive AI governance requires addressing non-model capability gains"
_ICML.cc/2026/Position_Paper_Track — ICML 2026 Position Paper Track regular_

### Official Review · Reviewer_AaXp · 2026-03-12

**Significance:** 3
**Argument Clarity:** 3
**Rating:** 4
**Confidence:** 4

**Questions:**

see above weaknesses

**Alternative Views Section:**

Yes

**Compliance With Llm Reviewing Policy A Conservative:**

Affirmed.

**Discussion Potential:**

3

**Final Justification:**

Some of my concerns have been addressed.

Regarding the first issue, while the authors have provided relevant empirical examples in Section 3.1, I believe that presenting these cases in a dedicated table, showing the specific examples alongside their corresponding non-model gain categories, would more effectively support the paper's arguments and enhance readability for the readers.

I keep my positive score.

**Paper Summary:**

This paper examines the limitations of "model-level governance"—the current dominant paradigm in AI safety that focuses on evaluating and mitigating risks at the level of the base model. The authors argue that as AI capabilities increasingly derive from "non-model gains" (improvements achieved after a model's initial training), model-level governance becomes insufficient. They categorize these gains into three primary types: inference gain (scaling compute at inference time), systems gain (enhancements through tools, scaffolding, and multi-agent setups), and asset gain (integration with proprietary data, hardware, or expertise). The paper also identifies emerging challenges like embodiment, continual learning, and diffusion effects. To address these shortcomings, the authors propose a diversified governance portfolio that complements model-level approaches with enhanced versions of it (e.g., forecasting, post-deployment monitoring) and novel strategies like system, entity, agent, and cloud governance, as well as societal resilience. The paper concludes with a call to action for researchers and policymakers to better understand non-model gains and develop these complementary mechanisms.

**Position:**

Yes

**Position In Title:**

Yes

**Related Work:**

3

**Strengths And Weaknesses:**

Strengths
The taxonomy of non-model gains (inference, systems, asset) is well-defined, clearly explained with concrete, recent examples (e.g., OpenAI o1, DeepMind's Big Sleep, government partnerships). This provides a solid foundation for the subsequent discussion on governance.
The paper looks beyond immediate trends to identify emerging challenges like embodiment, continual learning, and diffusion effects. Its core argument is proactive, advocating for diversification of governance strategies before model-level governance demonstrably fails, which is a valuable contribution to policy planning.
The paper is logically structured, moving from problem identification to analysis and then to potential solutions. The "Call to Action" section provides concrete, actionable steps for different stakeholders, enhancing its practical value.
The inclusion of a dedicated "Alternative Views" section strengthens the paper by demonstrating intellectual honesty. It directly addresses potential objections, allowing the authors to refine their position and show they have considered the complexity of the issue.

Weaknesses
The paper asserts that some non-model gains can match large pre-training scale-ups, but provides limited empirical quantification or case studies showing how large those gains are in representative settings. Adding worked examples or small empirical experiments (even simple benchmarking of a scaffold + inference compute vs a larger base model) would strengthen the claim.
Governance proposals remain high-level and would benefit from implementation detail. Proposals such as cloud-level monitoring, KYC for compute, or entity registration are discussed candidly, but the manuscript lacks fleshed-out protocols, threat models, or legal/operational feasibility analysis (costs, false-positive rates, privacy tradeoffs). Concrete toy designs or an appendix with an example audit flow would improve reproducibility of the policy analysis.

**Support:**

3

---

> ### Author Rebuttal · Authors · 2026-03-30
>
> Thank you for your thoughtful feedback.
>
> Regarding your comment: "The paper asserts that some non-model gains can match large pre-training scale-ups, but it provides limited empirical quantification or case studies showing how large those gains are in representative settings. Adding worked examples or small empirical experiments (even simple benchmarking of a scaffold + inference compute vs a larger base model) would strengthen the claim.":
>
> --> Please note that we do include empirical examples that do exactly what you propose under 3.1: “For example, a novel inference scaling technique called recursive self-aggregation enabled Qwen3-4B-Instruct-2507 to perform on par with larger reasoning models such as o3-mini (high) (Venkatraman et al., 2025). Additionally, DeepSeek-V3.2 outperformed the generally superior Gemini 3 on a series of prominent benchmarks by using between 1.5 and 2.5 times more tokens (DeepSeek-AI, 2025b).”
>
> Regarding your comment "Governance proposals remain high-level and would benefit from implementation detail. Proposals such as cloud-level monitoring, KYC for compute, or entity registration are discussed candidly, but the manuscript lacks fleshed-out protocols, threat models, or legal/operational feasibility analysis (costs, false-positive rates, privacy tradeoffs). Concrete toy designs or an appendix with an example audit flow would improve reproducibility of the policy analysis.":
>
> --> We strongly agree that fleshed-out protocols, threat models, and feasibility analyses are critical next steps. We point to those in our Call to Action. As a position paper, our primary contribution is establishing the necessity of non-model governance and mapping the capability taxonomy. Developing granular, legally robust implementation protocols (e.g., privacy trade-offs for cloud monitoring) requires dedicated, separate research efforts, which is precisely why we highlight these as priorities in our Call to Action section.

---

> > ### Author Rebuttal · Reviewer_AaXp · 2026-04-03
> >
> > Thanks for the response.
> >
> > Some of my concerns have been addressed.
> >
> > Regarding the first issue, while the authors have provided relevant empirical examples in Section 3.1, I believe that presenting these cases in a dedicated table, showing the specific examples alongside their corresponding non-model gain categories, would more effectively support the paper's arguments and enhance readability for the readers.
> >
> > I keep my positive score.

---

### Official Review · Reviewer_Jsf2 · 2026-03-13

**Significance:** 3
**Argument Clarity:** 3
**Rating:** 5
**Confidence:** 3

**Questions:**

- (Q1) The word \emph{governance} in the paper generally refers to a \emph{risk-oriented} approaches, i.e., to control models from risky outputs or actions. By contrast, there is \emph{management}, which refers to \emph{performance-oriented} approaches, i.e., to schedule and utilize models efficiently and appropriately. Does the authors feel it necessary to separate "management" from "governance", and make the distinction between these two concepts more explicit in the main text?

- (Q2) Compared to the terms "model-level" and "non-model" used in the article, I feel that the strategies described are better aligned with the concepts like "provider/owner-level" and "user/modifier-level". To be more specific, the capability profiles generated bon training data and the model itself correspond to “provider-level” governance, with enhanced model-level governance (such as improved elicitation standards, partnerships, etc.) presented as efforts that providers can further pursue, and the "non-model governance" introduced in this article generally refers to the profiles that incorporate information from system providers, entity-deployment organizations, agent infrastructures, cloud service providers as well as social users, who are either model users or model modifiers. Could the authors comment on and compare my understanding, have I missed some important claims in the paper?

- (Q3) What about model statistical specification proposed in learnware paradigm for model governance? (like Learnware of Language Models: Specialized Small Language Models Can Do Big; Learnware: Small Models Do Big)

**Alternative Views Section:**

Yes

**Compliance With Llm Reviewing Policy A Conservative:**

Affirmed.

**Discussion Potential:**

4

**Paper Summary:**

This position paper argues that the prevailing AI governance frameworks relying on model-level profiles which is primarily a function of the compute and data used in training process are becoming increasingly ineffective as capability progress is increasingly driven by non-model gains which refer to the improvements of model capabilities that are independent from advances in the base model.

The authors categorize these gains into inference gain (test-time compute scaling such as thought chain), systems gain (post-training enhancements such as scaffolds), and asset gain (access to restricted assets) and several future gain types. The authors then advocates for governance strategies not only including enhancements on model-level governance approaches such as cooperating with authorities, post-deployment monitoring and forecasting, but also non-model governance approaches that extending from model-level profiles to  encompass system, entity, agent, and cloud governance, supplemented by societal resilience. Finally, the authors call for improved metrics to track non-model gains, enhanced post-deployment monitoring, and broader evaluation of governance mechanisms to adapt to this transforming risk landscape.

**Position:**

Yes

**Position In Title:**

Yes

**Related Work:**

3

**Strengths And Weaknesses:**

- Strengths: The position stated in this paper is definitely on an important machine learning topic, that is, the governance of the frontier AI models. The position is clearly represented and supported with appropriate evidence and reasoning, which can inspire constructive discussion. The related works and alternative views are properly cited and discussed in main text. While the reviewer suggests further explanations of some concepts and viewpoints, the paper is of good quality and the reviewer votes for an acceptance.

- Weaknesses:
  - (W1) The definition and scope of "AI governance" \emph{in this paper} needs to be clarified. For example, does the authors intend to discuss "governing frontier AI models" or "governing each AI model" or "controlling the risks of the entire AI ecosystem"? If it is the latter, does the authors intend to obtain such a goal by "governing frontier AI models" or "governing each AI model" or others, and why not the other options are not covered in this paper? Some claims in the paper seems to rely on a specific claim above, for example, line 157-159 in left-half of page 3, the authors said that "model-level governance requires that powerful proprietary models maintain a lead over open-weight models", which may connote that it is hard to govern open-weight models.

  - (W2) Several claims in the paper needs to be elaborated to avoid possible mis-understandings. For example, again line 157-159 in left-half of page 3 as well as line 172-174 in left-half of page 4, these statements both possibly lead to a serious misunderstanding that "as long as models can obtain strong capabilities via mechanisms with small resource such as long thought chain or systematic organizations, then the risk becomes hard to control". The logics behind these statements should be made more comprehensive, since "uncontrolled risks" do not come from "strong capabilities", but come from the "unplanned / uncontrolled capabilities". Therefore, it would be better if the authors could provide some further discussions to make a clearer explanation. I've noticed that in line 319-321 in right-half of page 6, the authors state that "it struggles to address risks arising from open-weight models, decentralised development, or diffuse ecosystems where responsibility is fragmented". Can this statement serve as an explanation of the former two statements?

  - (W3, minor) In line 338-343 in left-half of page 7, the sentence immediately following the bolded sentence introduce a turn, so the meaning after the turn should also be emphasized as bolded.

**Support:**

3

---

> ### Author Rebuttal · Authors · 2026-03-30
>
> Thank you for your thoughtful feedback.
>
> W1:
>
> Our paper is about governing risks from the dangerous capabilities of frontier AI. This is distinct from "governing models", which is the dominant and flawed approach, as we explain throughout Section 2.  You're right to observe that 157-159 suggests open models are harder to govern. This makes model-level governance ("governing models") difficult - some of our "beyond model-level governance" suggestions address this (eg cloud governance).
>
> We will clarify what we mean by "AI governance" to avoid misunderstanding.
>
> W2:
>
> Yes, the assumption behind those arguments is indeed that if powerful capabilities are more broadly accessible to (unregulated) actors, there is a greater risk of misuse. We will clarify this assumption as you suggest.
>
> W3:
>
> Agreed - we will revise to make sure the emboldened sentence captures the entire paragraph argument.
>
> Q1:
>
> We do think of governance as risk/safety-focused (i.e. the common usage within AI safety community) and we acknowledge that in other sources, governance is sometimes used in a broader, managerial context. We’ll clarify our meaning in the definition paragraph (third para under Section 2).
>
>
> Q2:
>
> Your categorisation is also an important one, with both organising by where the capability gains originate (model vs non-model) and who the actor is (provider vs. user/modifier) providing valid lenses. We use model/non-model because we feel it captures more accurately the pain points in current governance efforts. Our non-model gains (inference, systems, restricted assets, etc) cannot be mitigated by simply shifting the current AI governance measures from provider to downstream user/modifier, even if that is a part of it (e.g. some but not all of the systems governance measures we describe are about replicating frontier safety on the system level). Rather, they demand a different kind of governance. Agent governance is an example of this: it's not about shifting responsibility from provider to a downstream actor, it's about rethinking the governance infrastructure altogether (e.g. do we need agent identification or protocols for inter-agent communication). Some of our 'beyond model-level governance' strategies are in fact aimed at providers (e.g. entity governance may also be relevant for frontier model providers - entity governance shifts the perspective from the model to the entity, not from entity A to entity B).
>
> Q3:
>
> Thank you for this pointer! Can you say more about how this paper is relevant for our argument? Are you thinking of learnware as a risk vector (a system-level gain?) or a governance strategy (how?).

---

### Official Review · Reviewer_4kaL · 2026-03-23

**Significance:** 3
**Argument Clarity:** 4
**Rating:** 5
**Confidence:** 3

**Questions:**

I was curious about the current practices in US, China, or India. I wonder how AI is currently governed in these markets and the recent trend for their AI policies.

**Alternative Views Section:**

Yes

**Compliance With Llm Reviewing Policy A Conservative:**

Affirmed.

**Discussion Potential:**

3

**Final Justification:**

I appreciate the authors' response. These are fundamental questions that need more proactive thoughts than a "solution". The authors' response has partially addressed my questions and contributed positively to my perception of this work.

I have read through the reviews and discussions, which seem generally on the positive side.

**Paper Summary:**

This paper presents an argument that the emergence of substantial non-model capability gains calls for corresponding responses in the AI government model. The paper discusses substantial non-model capability gains, specifically, inference gain (scaling compute at test-time), systems gain (post-training enhancements such as scaffolds), and asset gain (enhancing a model with restricted assets). The paper also discusses paths for potential solutions of beyond model-level governance, such as system governance, entity governance, agent governance, cloud governance, and societal resilience.

**Position:**

Yes

**Position In Title:**

Yes

**Related Work:**

2

**Strengths And Weaknesses:**

# +

- The paper is well-written. The logic is clear, the argument is substantiated, alternative views and potential solutions are presented and discussed.

- The topic is timely, and the discussions could be helpful to the community.

- Evidence is provided at appropriate places, and the paper is rich in cases and examples.

---

# -

* As much as it is timely, it can also be transient. It is hard to predict whether these factors will be as important in a longer term, which could have very different implications for the governance model.

* Besides, the paper's discussion on the governance model can benefit from more comparisons to current practices in US, China, or India. In these regions with large markets and where AI development is most active, their governance model and future path may be followed.

**Support:**

3

---

> ### Author Rebuttal · Authors · 2026-03-30
>
> Thank you for your thoughtful feedback.
>
> 1/ On the first weakness:
>
> While specific techniques (e.g. particular types of agentic scaffolding or methods of inference scaling) will inevitably evolve, the structural trend (i.e. the decoupling of a system’s capability from the base model/pre-training) seems durable. This follows from the diminishing returns on pre-training scaling and its exponentially increasing costs (3.5x/year, according to Epoch) plus the undesirability of extremely long training runs (Epoch: Frontier training runs will likely stop getting longer by around 2027). The paper Meek Models Shall Inherit the Earth (Gundlach et al, 2025) explains how this hollows out the advantages associated with ever-larger pre-training runs.
>
> We don’t make statements about which of our non-model gains will be the most important and the paper accounts for the long-term trajectory as we explicitly discuss future non-model vectors such as embodiment, continual learning, and diffusion effects in addition to discussing already visible trends. The diversification of the governance toolkit is useful regardless of the precise vectors of capability gain, and as explained in alternative views, "Our argument is that preparing governance mechanisms beyond the model level is prudent given the trajectory of non-model gains. Waiting instead until model-level governance has demonstrably failed risks leaving the governance ecosystem unprepared.”
>
> Given the breadth of capability vectors and governance models covered, and the emphasis on the overarching structural trend (the relative decline of pre-training as a driver of capabilities, as highlighted in 2 and supported with ample empirical evidence, and reiterated in 5): we respectfully disagree with the suggestion that we are over-indexing on a potentially transient model of capability gain. That being said, we will add some clarifying language to summarise the argument in the bullets above at the beginning of Section 3.
>
> 2/ On the second weakness:
>
> Due to space constraints, a detailed comparative analysis of regional policies is difficult to accommodate. However, we agree with the premise that characterising governance models of major powers is useful. Besides the EU, which has the most prominent and comprehensive frontier AI regulation, we already make some connections between US policy measures and the model-level paradigm (end of third paragraph, section 2). We will leverage extra space to make some more connections (e.g. US Executive Order and State Bills rely heavily on pre-training compute thresholds, which will become less effective due to non-model gains). China has a lot of rules and standards regarding pre-deployment safety assessments which also fit within our argument, though they are typically less focused on *frontier* AI governance which is the scope of our paper. Since our submission, China has released its 15th 5-year plan, which contains a substantive section on AI, addressing some safety and governance topics. This includes both model and non-model governance measures, which we can briefly address in the next iteration of the paper.

---

> > ### Author Rebuttal · Reviewer_4kaL · 2026-04-03
> >
> > I appreciate the authors' response.  These are fundamental questions that need more proactive thoughts than a "solution". The authors' response has partially addressed my questions and **contribute positively to my perception of this work. I have increased my score 3->5.** I have read through the reviews and discussions, which seem generally on the positive side.  Good effort and good luck.

---

### Decision · Program_Chairs · 2026-04-30

**Decision:**

Accept (regular)

**Comment:**

The paper presents a timely and relevant topic that will inspire productive discussion and future work in frontier AI model governance. The paper presents compelling evidence and example cases with a clear call to action.